# Decay Incidence and Quality Changes of Film Packaged 'Simeto' Mandarins Treated with Sodium Bicarbonate

**Salvatore D'Aquino** [1,*]**, Maria Concetta Strano** [2] **, Alessandra Gentile** [3] **and Amedeo Palma** [1]

1　Institute of Sciences of Food Production, National Research Council, Traversa La Crucca 3, 07100 Sassari, Italy; amedeo.palma@cnr.it

2　Council for Agricultural Research and Economics, Research Centre for Olive, Fruit and Citrus Crops, Corso Savoia 190, 95024 Acireale, CT, Italy; mariaconcetta.strano@crea.gov.it

3　Department of Agriculture, Food and Environment, University of Catania, 95123 Catania, Italy; gentilea@unict.it

*　Correspondence: salvatore.daquino@cnr.it

**Abstract:** Not rinsing sodium bicarbonate (SBC) treated fruit with freshwater can reduce post-harvest decay, but it can also be phytotoxic to peel tissues. Film packaging delays the ageing of peel, due to the high in-package humidity, but this also stimulates the growth of pathogens. Thus, as stand-alone treatments, both SBC and film packaging present advantages, but also drawbacks. In this study, SBC phytotoxicity was effectively mitigated when 'Simeto' mandarins, subjected to a 2 min dip treatment in a 2% SBC solution, were packaged using Omni film (highly permeable to water vapor and gases) or Coralife SWAF 400 film (with a low permeability to water vapor, but moderately permeable to gases). In particular, the combination Coralife SWAF 400 film allowed the fruit to be stored for 7 d at 5 °C, or 14 d at 20 °C, with negligible changes in overall appearance, almost no loss caused by decay, and an average weight loss of 1.3%. The in-package air composition, similar to air in Omni packages, and with an average between 5 kPa $CO_2$ and 16 kPa $O_2$ in Coralife SWAF 400 packages, slightly affected the sensory and chemical qualities. Combining SBC with film packaging is a feasible method to prolong the post-harvest life of citrus fruit, and control post-harvest diseases, while avoiding the use of synthetic fungicides.

**Keywords:** cold storage; citrus fruit; packaging; modified atmosphere; fruit quality; shelf-life

## 1. Introduction

Refrigeration is considered the most effective technological means to prolong the post-harvest life of fresh fruit, owing to its capacity to slow down the overall metabolic activity, the free energy of water, and consequently, transpiration and the growth of pathogens. Regarding citrus fruit intended for fresh markets, an efficient control of the ambient humidity is equally as, or even more important, than low temperature [1,2]. After harvest, citrus fruit, just as other non-climacteric fruit, undergo slight changes in the chemical composition of the edible portion, even when stored at room temperature. This is because of the degradation rate of respirable substrates, which is quite slow and dependent on the respiratory activity of the segments, several times lower than that of the peel [3]. In contrast, improper control of the environmental humidity, regardless of the storage temperature (particularly in mandarins and other cultivars with loose adhering rind), hastens water loss and tissue softening, decreases the level of insoluble pectins and increases that of soluble ones, while triggering other physiological processes attributed to senescence [4]. Excessive water loss may also alter the flavor and taste, thus, shortening the shelf-life and lowering the commercial value [5]. Improper management of temperature and humidity during transportation, and in retail environments and consumers' homes can rapidly induce fruit shriveling and stem-end rind breakdown [6–10].

A very effective way to maintain humidity in non-refrigerated conditions is modified atmosphere packaging (MAP). MAP can almost completely inhibit water loss, and prolong citrus fruit freshness for several weeks in unrefrigerated conditions, although restricted gas exchanges and high in-package humidity can trigger anaerobic processes and the growth of pathogens [11].

Anaerobiosis occurs when gas exchange in packaging is not sufficient to efficiently control the levels of in-package $CO_2$ and $O_2$ altered by the respiration of the packaged produce; the consequent $O_2$ depletion and $CO_2$ increase may lead to fermentative processes, and the build-up of off-flavors and off-odors [12]. As the permeability of most conventional packaging polymers does not match fresh fruit and vegetables requirements, macro- or micro-perforations are frequently deployed to enhance gas exchange [13]. High in-package humidity favors decay development, especially when fruit are stored at warm temperatures, and condensation forms within the packages [12].

Post-harvest treatments with synthetic fungicides can effectively reduce the risk of decay, especially when fruit are directly delivered for consumption. Nevertheless, stringent regulations in many countries, aimed at reducing the impact of pesticides on human health and the environment, limit their use and make them less popular with consumers, who prefer to buy chemical-free produce, even if at a higher price. Yet, the continuous use of a restricted number of synthetic fungicides has caused the development of resistant strains of fungi-causing decay. As a result, in the last decades several treatments, alternatives to synthetic fungicides, were developed and combined with the synthetic fungicides, such as the incorporation of edible coatings, or used as stand-alone treatments, in order to improve and/or reduce fungicide activity, prevent the risk of resistance, and reduce the health risks posed by continued exposure to several chemical compounds [14–18]. Immersion in hot water (HWD) and the use of inorganic salts, such as sodium bicarbonate (SBC) or sodium carbonate (SC), have reduced the development of pathogens in citrus fruit, although less efficiently than synthetic fungicides [19–22]. HWD is active against germinated conidia, but less so against dry conidia; its efficacy increases when the temperature is raised, and the duration of the treatment is extended, but the best combination, which could completely inhibit pathogens, becomes phytotoxic to fruit tissue [19,23]. In addition, as HWD leaves no residue on the fruit surface, ungerminated conidia, and those landing on the fruit surface subsequent to treatment, can initiate new infections.

SBC is fungistatic and shows a moderate inhibitory activity at room temperature on most fungi-causing decay; its efficacy increases with temperature and concentration, but as the solution temperature increases, its solubility in water decreases and the release of $CO_2$ in the air occurs at high rates [20]. For this reason, the temperature of the SBC solution generally does not exceed 45 °C [24]. At commercial level, the potential SBC activity is limited by the practice of rinsing the fruit with fresh water after the treatment, to prevent the deposition of the salts from brushes, packing belts, and sorting equipment, and the staining and desiccation of fruit rind [24]. Yet, SBC can cause a partial removal of the epicuticular waxes, which increases water loss, especially when the fruit is stored at ambient conditions and/or low humidity [25]. Storing the fruit in environments with high humidity mitigates the phytotoxic effects of SBC, and extends the protective effect of SBC if fruit are not rinsed. Recently, D'Aquino et al. [9] show that the combination of the individual bagging of lemons treated with SC, but not then rinsed with fresh water, led to an effective control of decay, without any detrimental effect on appearance and weight loss for a long time.

In this study, film packaging was combined with HWD or SBC to delay quality loss and the decay development of 'Simeto' mandarins stored for 7 d at 5 °C, to simulate refrigerated transportation, plus 14 d at 20 °C, to simulate the number of potential days that elapse between the beginning of market display, and when the fruit are eaten at home. The objective was to prolong the fruit overall quality, and to reduce microbiological spoilage, by designing a suitable modified-atmosphere packaging, relying on packaging

material available on the market and exploiting the potential of SBC as an alternative to synthetic fungicide.

## 2. Materials and Methods

### 2.1. Plant Material, Treatments and Storage Conditions

'Simeto' mandarins, a hybrid of the 'Miho' satsuma and the 'Avana apireno' mandarin were harvested from a commercial orchard located in Sassari. Immediately after harvest, the fruit were delivered to the laboratory, where a total of 2700 fruit, free of visual defects, were individually weighed and divided into three groups. The first group (CWD) were immersed in water at 20 °C for 2 min, and served as the control; the second group was immersed in water heated at 45 °C for 2 min (HWD); the third group was immersed in a 2% sodium bicarbonate solution (SBC) heated at 45 °C for 2 min (Eurospin, Mantova, Italy); for simplicity these are indicated as fungicidal treatments. Three hours later, after natural drying, all fruit were placed in carton boxes (0.30 × 0.20 × 0.11 m) in numbers of 10 per box, for a total of 270 boxes, 90 boxes per treatment. The fruit of each treatment were divided into three sub-groups of 30 boxes each. The fruit of the first subgroup were left un-wrapped within the boxes, while those of the second and third subgroups were sealed within 0.30 × 0.28 m bags, made with a 15 μm thick polyvinylchloride film (Omni, Erre-Ci-A, Monza, Italy) or a 20 μm polypropylene film (Coralife SWAF 400, Corapack SRL, Como, Italy), whose characteristics are reported in Table 1.

**Table 1.** Barrier properties of the plastic films used and method used for determination.

| Film | $O_2$ Permeance [1] $(\mu mol\ s^{-1}\ m^{-2}\ kPa^{-1})$ | $CO_2$ Permeance $(\mu mol\ s^{-1}\ m^{-2}\ kPa^{-1})$ | Water Vapor Transmission Rate $(\mu g\ s^{-1}\ m^{-2})$ |
|---|---|---|---|
| Omni | 0.145 (ASTM D-1434) | 1.048 (ASTM 1434) | 8000 (ASTM E-96) |
| Coralene SWAF 400 | 0.011 (ASTM D-3985) | 0.044 [2] | 81.2 (ASTM E-96) |
| Coralife SWAF 400 [3] | 0.181 (ASTM D-3985) | 0.175 (COV-E68) | 83.4 (ASTM F 1249-90) |

[1] Data, provided by the manufacturers, were transformed in SI units; [2] not provided by manufacturer; calculated considering a $CO_2/O_2$ equal to 4; [3] calculated by adding Coralene SWAF permeance to $CO_2$ and $O_2$, and transmission rate to water vapor the effect of 1250 laser perforations per meter square. Diameter of each laser perforation equal to 80 micron. $O_2$ permeability thorough perforations was considered 1.30 times that of $CO_2$.

Before packaging, a data logger (LogTag Humidity & Temperature Recorder, Auckland, New Zealand) was placed inside three bags of each treatment to measure the in-package temperature and humidity at 24 h intervals. Three more data loggers were placed in different sites of the storage room to monitor the environmental temperature and humidity. The average weight of each fruit was 195 ± 12.6 g, for a total weight of about 2 kg per box. Fruits were stored at 5 °C and 90% RH (CS) for 7 d, plus 1 or 2 weeks at 20 °C and 55–60% RH to simulate marketing conditions and the time fruit are held at home before consumption (SMC).

At week intervals, 10 bags of each treatment combinations were used for analyses and assessments.

### 2.2. Respiratory Activity, in-Package $CO_2$ and $O_2$ Partial Pressure, $O_{2epp}$ and $CO_{2epp}$, $RCO_2$, in Package Temperature and Humidity

In-package $CO_2$ and $O_2$ partial pressure were determined using a hand-held analyzer (Check Point, PBI-Dansensor, Italia, Milan, Italy) for combined measurements of oxygen and carbon dioxide. The needle was inserted through a strip of an adhesive electrical tape stuck on the film, to avoid making tears with the needle when sampling.

Respiratory activity, $CO_2$ ($CO_{2epp}$), and $O_2$ ($O_{2epp}$) endogenous partial pressure were determined at harvest, and subsequently at week intervals, immediately after film removal (t1), and after five hours (t2) to let endogenous gases re-equilibrate with the atmosphere.

For respiration, five fruit of each treatment were individually placed in 1 L jars, fitted with two silicon septa, and closed for 1 h prior to $CO_2$ determination. At sampling time, the headspace air was mixed for 1 min by an electrical fan fixed inside the jar. $CO_2$ concentration was determined using a combined $CO_2/O_2$ analyzer (Combi Check 9800-1, PBI-Dansensor A/S, Rinsted, Denmark); respiration rate, just as carbon dioxide release, was expressed as mL $CO_2$ $Kg^{-1}$ $s^{-1}$ [1]. Based on respiration activity and the film barrier properties, the expected in-package partial pressure of $CO_2$ and $O_2$ were calculated (Table 2).

**Table 2.** Expected in-package $CO_2$ and $O_2$ partial pressure. Calculations were carried out assuming a respiration rate of 3.5 (5 °C) and 16 mL $CO_2$ $kg^{-1}$ $h^{-1}$ (20 °C), and a film package surface equal to 0.36 $m^2$. The ratio of $CO_2$ released to $O_2$ consumed was considered equal to 1.

| Film | Expected in-Package $CO_2$ Partial Pressure (kPa) | | Expected in-Package $O_2$ Partial Pressure (kPa) | |
|---|---|---|---|---|
| | Storage at 5 °C | Storage at 20 °C | Storage at 5 °C | Storage at 20 °C |
| Omni | 0.23 | 1.05 | 19.3 | 13.4 |
| Coralife SWAF 400 | 1.38 | 6.29 | 19.7 | 14.9 |

Five fruit were also used to analyze the internal atmosphere; after submerging the fruits in water, a 1 mL sample of the internal atmosphere was withdrawn by inserting the needle of a syringe from the stylar end. $CO_{2epp}$ and $O_{2epp}$ were determined by injecting the gas sample into a Varian 3300 gas chromatograph equipped with a joined 2 m length × 6.28 mm i.d. CTR-I (Alltech, Milan, Italy), and a TCD detector (test conditions: ambient temperature for injection, detector, and oven; filament temperature 100 °C, helium carrier gas at a flow rate of 40 mL $min^{-1}$).

Modifying the formula reported by Trout et al. [26] [$(CO_{2epp} - CO_{2amb})/Resp$], where $CO_{2amb}$ was the ambient $CO_2$ partial pressure, the resistance to gas diffusion of $CO_2$ ($RCO_2$), expressed as kPa (mL $kg^{-1}$ $h^{-1})^{-1}$, was calculated.

### 2.3. Decay, Weight Loss, Overall Appearance and Marketable Fruit

Decay incidence (expressed as percentage), overall appearance, and weight loss were evaluated after 1, 2, or 3 weeks on 10 boxes of each treatments' combination by three trained technicians. The number of total fruit for each treatment combination was 100 for decay incidence; the remaining sound fruit, equal to 100 minus the number of decayed fruit, were used for overall appearance evaluation, and weight loss determination.

Overall appearance was judged on a scale ranging from 1 to 9 (1 = fruit very aged; 5 = fruit aged but still saleable; 7 = fruit still fresh; 9 = fruit as fresh as at harvest). Overall appearance was calculated as the weighted average of the score of each individual fruit. The overall appearance of marketable fruit was calculated by computing only the score of fruit rated 5 or higher.

The percentage of marketable fruit compared to the initial number of fruit, was calculated according to the Equation (1):

$$MFN = (Ni - Nd - Nu)/Ni \times 100 \tag{1}$$

where MFN is the percentage of fruit scored 5 or higher; Ni is the initial number of fruit (100); Nd the number of decayed fruit; and Nu the number of fruit scored less than 5.

At each sampling date, fruits were weighed, and the weight loss was calculated as the percentage of weight reduction with respect to the initial weight according to the Equation (2):

$$WL = 100 \times (Wi - Wt)/Wi \tag{2}$$

where WL is the weight loss (%); Wi the initial weight; and Wt the weight at time t.

The percentage of marketable fruit, as a percentage of the initial weight, was calculated according to the Equation (3):

$$MFP = MFN \times (WL/100) \tag{3}$$

where MFP is weight percentage of marketable fruit compared to initial weight; MFN is marketable fruit as number percentage of the initial number; and WL is weight loss as percentage of the initial weight.

### 2.4. Firmness and Color

Firmness assessments were carried out by a testing machine (Mod. DO-FB 0.5 TS-Zwick-Roell, Ulm, Germany). For the penetration test, the highest resistance (N) opposed to the penetration of a 2 mm diameter flat-faced cylindrical plunger, to a depth of 8 mm and moving at a speed of 1.7 mm s$^{-1}$, was recorded. The deformation test was accomplished by measuring the reduction of the equatorial diameter in mm after a 1 kg force was applied to a 5 cm diameter plate, moving at an initial speed of 0.83 mm s$^{-1}$.

The color of the peel was measured with a Minolta CR-300 colorimeter (Minolta, Osaka, Japan) according to the Commission Internationale de l'Eclairage (CIE) (L\*, a\*, b\*) color scale [27], and the chroma (color saturation, C = [a\*$^2$ + b\*$^2$]$^{1/2}$) and hue angle (H$°$ = tan$^{-1}$ [b\*/a\*]) were calculated.

For both determinations, ten fruit per treatment were used.

### 2.5. Chemical Analysis

The juice of three replications (10 fruit × replication) for each treatment combination, obtained by squeezing the fruit with a commercial juicer, was used for chemical analyses, after centrifugation (Centurion Scientific Ltd., West Sussex, England) at 13,000× $g$ for 15 min and filtration of the supernatant through a 0.45 μm acetate cellulose filter. All analyses were performed in triplicates.

The pH of the juice was determined using a potentiometric titrator (Metrom 720 SM Tritino, Swiss).

The percentage of total soluble solids (TSS) was measured by a digital refractometer (PR-101, Atago, Japan).

Citric acid and ascorbic acid were determined with a Merck-Hitachi (Tokyo, Japan.) liquid chromatograph (HPLC) with an L-7455 photodiode (DAD) detector, D-7000 system manager, L7200 autosampler, and L-7100 pumps. The procedure followed was described by Yuan and Chen [28], and Chinnici et al. [29], using a Bio-Rad cation guard column and a Bio-Rad Aminex HPX-87H hydrogen form cation exchange resin-based column (300 mm × 7.8 mm i.d.) at 40 °C. The mobile phase consisted of 0.005 M sulfuric acid aqueous solution, and the samples were isocratically separated at 0.6 mL min$^{-1}$. Peaks of citric acid and ascorbic acid were measured at wavelengths of 210 and 245 nm respectively, and were identified by comparing retention times with those of standards. Quantification was carried out using external standards.

Total phenolic content was analyzed according to the Folin–Ciocalteu colorimetric method [30]. Total phenols were expressed as gallic acid equivalent.

Antioxidant activity was assessed using the free radical DPPH, according to Bonded et al. [31]. The mixture, containing 3 mL of a methanol solution of 0.16 mM DPPH and 100 μL of diluted sample (1/10), was allowed to react for 15 min in a cuvette. The decrease in absorbance at 515 nm of DPPH solution added to the sample was measured at 20 °C after 15 min, and the antioxidant activity was expressed as mmol Trolox equivalent L$^{-1}$ (TEAC).

For ethanol and acetaldehyde determination, 2 mL of juice were placed in a 10 mL headspace vial and incubated for 2 h in a shaking bath at 60 °C. Then, a 1 mL headspace gas sample was injected into a gas chromatograph (Agilent 6890) fitted with a flame ionization detector (FID), and equipped with a 5% Carbowax 20 M on 80/120 Carbograph 1 AW packed column. Run conditions were: N$_2$ as carrier gas at 30 mL min$^{-1}$; injector at 130 °C;

oven 80 °C; detector at 150 °C. Ethanol and acetaldehyde were quantified by a comparison of peak area versus concentration of a calibration curve obtained from pure analytical standards subjected to the same analytical conditions [32].

### 2.6. Sensory Evaluation

Sensory evaluation was performed by five trained technicians. Nine fruit per treatment (3 fruit from each package), at each sampling time, were peeled, divided into segments, and used for taste analysis. Each assessor ate 2–3 segments of each treatment and gave a score for sweetness, acidity, off-flavor, firmness (overall impression of hardness for the bites), and overall acceptability based on a 9-point scale, according to the intensity of the attributes. Overall acceptability was measured differently, for which a score of 5 to 9 indicated that the fruit would be acceptable to eat, while a score of 4 or less that the fruit would be not acceptable.

### 2.7. Statistical Analysis

Statistical analysis was performed using Statgraphics Centurion software (Herndon, VA, USA), version XV Professional statistical program. The experiment was designed as a complete randomized block design with 3 factorial arrangements of treatments: P (storage period) with 3 (the 3 inspections) or 4 levels when data were compared with those of harvest time; T (treatment), with 2 levels, indicating the control, the immersion in hot water, and the immersion in SBC solution; and F, with 2 levels (unwrapped, Omni film, and XX film).

As T did not affect juice chemical parameters, only the P × F interactions are presented. At each inspection time, a one-way analysis of variance (ANOVA) was carried out comparing T × F interactions, or the levels of F (chemicals analyses), and means separation was accomplished using the Duncan's multiple range test at $p < 0.05$.

## 3. Results

### 3.1. Respiratory Activity, in-Package $CO_2$ and $O_2$ Partial Pressure, $O_{2epp}$ and $CO_{2epp}$, $RCO_2$, in Package Temperature and Humidity

Respiration is affected by storage time and film packaging (data not shown). Initially, it is around 16 mL $CO_2$ kg$^{-1}$ h$^{-1}$, and drops to 4–5 mL $CO_2$ kg$^{-1}$ h$^{-1}$ when fruit are moved to 5 °C. In packaged fruit, the differences between the determinations at the time of film removal, and those performed following 5 h film removal are negligible (Figure 1A). In SMC, Coralife-packaged fruit shows a particularly high $CO_2$ flux immediately after film removal (about 70 and 90 mL $CO_2$ kg$^{-1}$ h$^{-1}$, after 1 or 2 weeks, respectively), which declines very rapidly after 5 h, with values not significantly different to unwrapped fruit. In Omni-packaged fruit, soon after film removal, $CO_2$ flux is about two-fold higher than unwrapped fruit, but declines slightly after 5 h (Figure 1A).

Both in-package $CO_2$ and in-package $O_2$ are affected by storage time and film wrapping, but not by treatments (data not shown).

During CS, $CO_2$ partial pressure is slightly higher than air in Omni packages, and averaged 1.5 kPa in Coralife packages. In-package $CO_2$ increases to about 0.55 kPa in Omni packages when fruit are moved to 20 °C; no remarkable change occurs thereafter. Inside Coralife packages, $CO_2$ is 4.66 kPa after 1 week at 20 °C, and increases to 5.7 kPa after 2 weeks (Figure 2B). In Omni packages, $O_2$ is constantly above 20 kPa both at 5 °C and at 20 °C, whereas in Coralife packages it is around 19 kPa at 5 °C, but decreases to about 17.1 kPa after 7 d of SMC, and to 15.8 kPa after 14 d (Figure 3B).

Initially, $CO_{2epp}$ partial pressure is 1.24 kPa; it drops to 0.2–0.5 kPa at the end of the 7 d of CS, with the highest and the lowest values detected in packaged fruit soon after film removal, and after 5 h, respectively (Figure 1D).

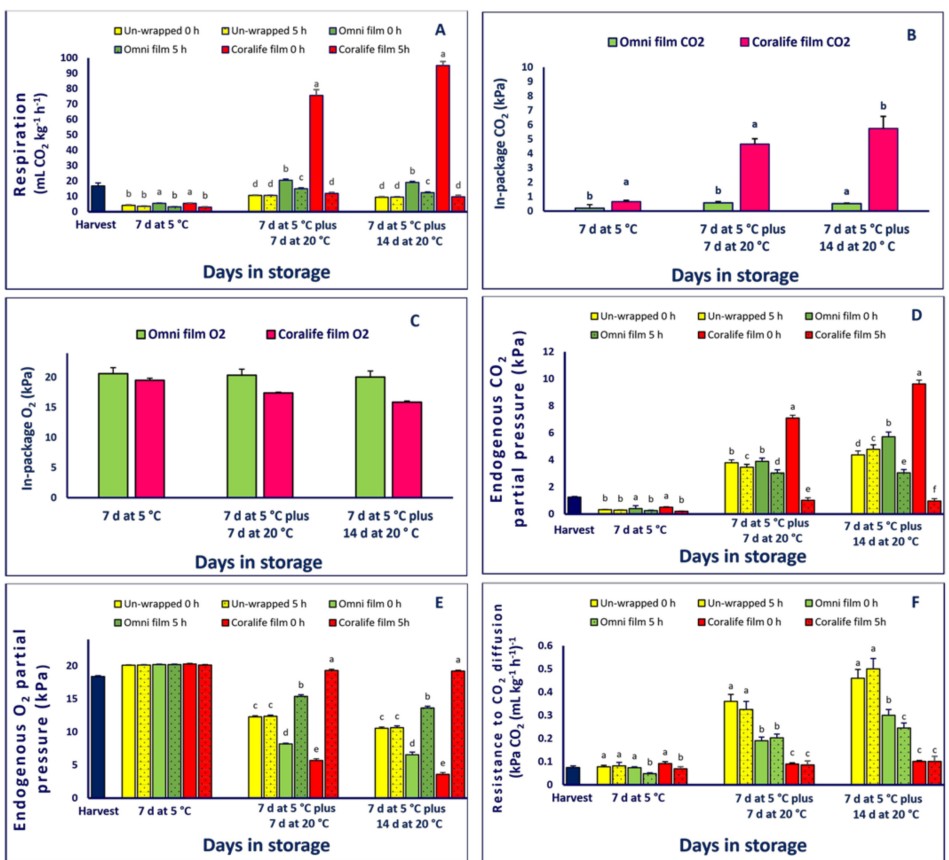

**Figure 1.** Respiration (**A**), in-package $CO_2$ (**B**) and $O_2$ (**C**), endogenous $CO_2$ (**D**), $O_2$ (**E**), and resistance to $CO_2$ diffusion(**F**) of 'Simeto' mandarins. Vertical bars represent the standard deviation (n = 5 for respiration; n = 10 for in-package $CO_2$ and $O_2$; n = 5 for endogenous $CO_2$, endogenous $O_2$, and resistance to $CO_2$ diffusion. For each storage time, means followed by unlike letters are significantly different at $p \leq 0.05$. Means separation was accomplished by Duncan's multiple range test.

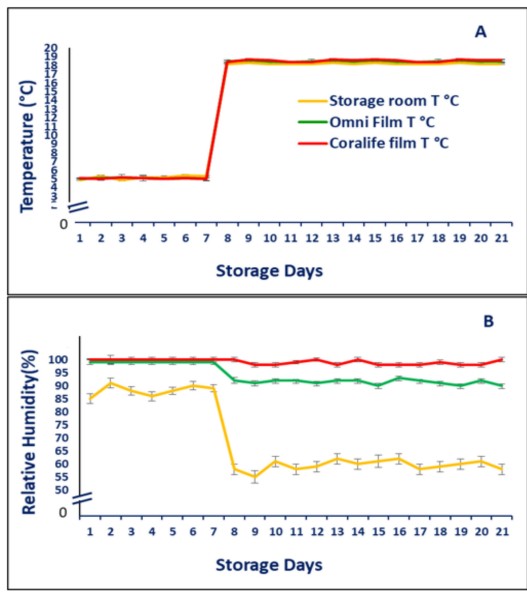

**Figure 2.** Average temperature (**A**) or humidity (**B**) of the storage room or the packages headspace. For each day, means followed by unlike letters are significantly different at $p \leq 0.05$. Means separation was accomplished using Duncan's multiple range test. Vertical bars are the standard deviations of three replications (n = 3).

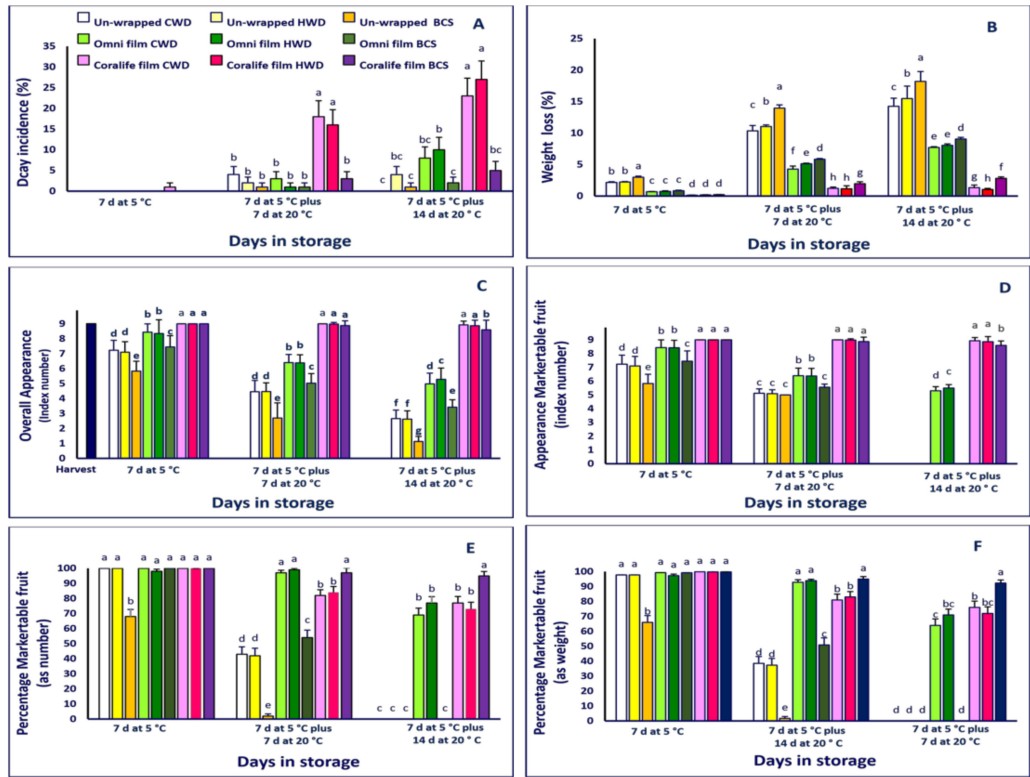

**Figure 3.** Decay incidence (**A**), weight loss (**B**), overall appearance (**C**), appearance of marketable fruit (**D**), percentage of marketable fruit as number (**E**), and weight (**F**) compared to harvest time of 'Simeto' mandarins. For each storage period, means followed by unlike letters are significantly different at $p \leq 0.05$. Means separation was accomplished using Duncan's multiple range test. Vertical bars are the standard deviations of ten replications (n = 10), each composed of 10 fruits minus the number rotten ones.

$CO_{2epp}$ increases in all treatments upon transfer to SMC. In unwrapped fruit, $CO_{2epp}$ is about 3.6 kPa after 7 d, and 4.6 kPa after 14 d (Figure 1D). In Omni-packaged fruit, $CO_{2epp}$ detected soon after film removal is not statistically different than that of unwrapped fruit determined after 7 d, but is slightly higher after 14 d. At day 7, as well at day 14, 5 h following film removal, $CO_{2epp}$ drops to about 3 kPa. In Coralife packages, $CO_{2epp}$ detected soon after film removal is 7.1 and 9.2 kPa, after 7 and 14 d, respectively; subsequently, 5 h after the film removal, $CO_{2epp}$ drops to about 1 kPa, with no significant difference between determinations done at day 7 or 14 (Figure 1D).

$O_{2epp}$ is 18.4 kPa at harvest and increases to about 20 kPa after 7 d at 5 °C, with no detectable difference among treatments. $O_{2epp}$ declines in all treatments upon transfer to SMC (Figure 1E). In unwrapped fruit, $O_{2epp}$ is 12.4 kPa after 7 d, and 10.7 kPa after 14 d; in Omni-packaged fruit, immediately after film removal, $O_{2epp}$ is 8.02 and 6.6 kPa after 7 and 14 d, respectively; it increases to 15.4 and 16.7 kPa after 7 and 14 d, respectively, when determined 5 h after film removal (Figure 1E). Coralife fruit shows the lowest $O_{2epp}$ at the time of film removal (5.7 and 3.6 kPa after 7 and 14 d, respectively), and the highest $O_{2epp}$ values after 5 h (19.3 and 19.1 at day 7 and day 14, respectively).

At harvest time, $RCO_2$ is 0.078 kPa (mL kg$^{-1}$ h$^{-1}$)$^{-1}$; after 7 d of CS at 5 °C, $RCO_2$ shows no appreciable changes. Upon transfer to SMC, $RCO_2$ increases in unwrapped fruit (4.6 and 6.5 fold higher at day 7 and day 14, respectively), and in Omni-packaged fruit (2.6 and 3.7 fold higher at day 7 and day 14, respectively). In Coralife-packaged fruit, $RCO_2$ slightly increases, with values never exceeding 0.1 kPa (mL kg$^{-1}$ h$^{-1}$)$^{-1}$ (Figure 1F). Changes in $RCO_2$ occurring 5 h after the film removal are negligible in both package treatments.

The ambient and the in-package humidity, as well as the temperature, changed significantly during storage (data not shown).

The ambient temperature during the 7 d of CS shows a swing of about ±0.5 °C around 4.9 °C; variations during the 14 d of SMC are narrower (19.8 °C–20.1 °C). In CS, as well as in SMC, differences in temperature between the in-package atmosphere and the storage room are negligible (Figure 2A).

The ambient RH is quite stable during the 7 d at 5 °C, averaging 85%, while fluctuating between 55 and 61% in SMC. The headspace RH of Omni (average 96%) and Coralife (average 100%) packages are quite stable while fruit were stored at 5 °C. Upon transfer to SMC, the RH of Omni packages decreases significantly and fluctuates between 85 and 92%, whereas in Coralife packages, no significant change occurs, with values always close to 100% (Figure 2B). Yet, upon fruit transfer to SMC, water condensation formed within all packages. However, while in Omni packages it completely disappeared within the next 24 h, condensation persisted in Coralife packages.

*3.2. Decay, Weight Loss, Overall Appearance and Marketable Fruit*

Decay is significantly affected by all experimental factors and their interactions (data not shown). At the end CS, fruit of all treatment combinations are sound, except for Omni-HWD (1% rotten fruit). Pathogens, especially Penicillium digitatum, start to grow when fruit are moved to SMC. In unpackaged fruit, differences among fungicidal treatments are never significant; losses increase after 7 d of SMC, but show no further increase during the next 7 d (Figure 3A). During the 14 d of SMC, differences among fungicidal treatments are significant in packaged fruit, with SBC showing very effective results in all treatment combinations; in contrast, HWD always shows no significant difference compared to the control treatment (CWD). In Omni packages, overall decay is quite low, with almost no loss until day 7; after 14 d, decay increases to 8% in CWD-Omni packages, and to 10% in HWD-Omni ones; negligible losses are detected in SBC-Omni packages (Figure 3A). Losses are dramatically higher in Coralife packages, especially in HWD (27% after 14 d) and CWD (23% after 14 d) treatments, but never exceeding 5% in BCS-Coralife packages (Figure 3A).

All experimental factors, and all their interactions, affect overall appearance (data not shown). Overall appearance declines at a very fast rate in all unpackaged fruit, at a slower rate in Omni-packages, and very slowly in Coralife ones (Figure 3B).

Consequently, overall appearance of marketable fruit (fruit scored 5 or more) is significantly affected by storage time, fungicidal treatments, and packages (Figure 3C). The score of unpackaged, marketable fruit declines rapidly even in cold storage, particularly in unpackaged SBC fruit (score = 6), which is close to the limit of marketability after 7 d of SMC, while no fruit is judged as marketable after 14 d of SMC. In Omni-SBC packages, the appearance of marketable fruit declines markedly even in CS; when fruit is moved to SMC, a marked reduction of freshness occurs in all Omni packages just after 7 d. After 14 d, only CWD and HWD Omni-packaged fruit are still marketable, while no Omni-BCS fruit is still marketable (Figure 3C). The appearance of Coralife-packaged fruit is slightly affected by storage time and fungicidal treatments, and, even in fruit treated with SBC, the score of marketable fruit after 14 d is still higher than 8 (Figure 3C).

All experimental factors affect weight loss, although at a different degree (data not shown). HWD enhances weight loss, but less than BCS, while both packaging films reduce weight loss, and mitigate the negative effect of HWD and BCS. However, Coralife film, with final losses ranging between 2.55% (BCS) and 1.56 (CWD), is much more effective than Omni film (Figure 3D).

The percentage of marketable fruit (PMN), as a number compared to the initial number of fruits, is affected by all experimental factors (data not shown). At the end of CS, PMN has declined markedly in unpackaged SBC fruit and Coralife-CWD fruit (70% and 85%, respectively), while in all other treatment combinations, PMN is still close to 100% (Figure 3E). As the storage time progresses, the combined negative effect of decay and loss of overall appearance drastically reduces the PMN; after the first 7 d of SMC, the highest

PMN values are detected in Omni-HWD, Omni-SBC, and Coralife-SBC treatments, but at the end of storage, due to the increase in decay of packaged fruit, PMN in Coralife-CWD and Coralife-HWD declines to 77% and 73, respectively, whereas in Coralife-SBC the PMN is still 95%. After 14 d of SMC, no unpackaged fruit, and no Omni-SBC fruit are marketable, due to the severe loss of appearance.

Regarding the PMW, the lowest values are detected in those treatments where the combined effect of decay incidence, decline of overall appearance, and weight loss are higher (Figure 3F).

### 3.3. Firmness and Color

Resistance to puncture is affected by the storage period, film packaging, and their interactions (data not shown). Compared to harvest time, no change occurs after the 7 d of CS in all treatments. Resistance to puncture increases significantly in all unpackaged fruit after 7 d of SMC; no further change occurs thereafter (Figure 4A). Resistance to puncture decreases over the 14 d of SMC in all packaged fruit, particularly in Coralife packages (Figure 4A).

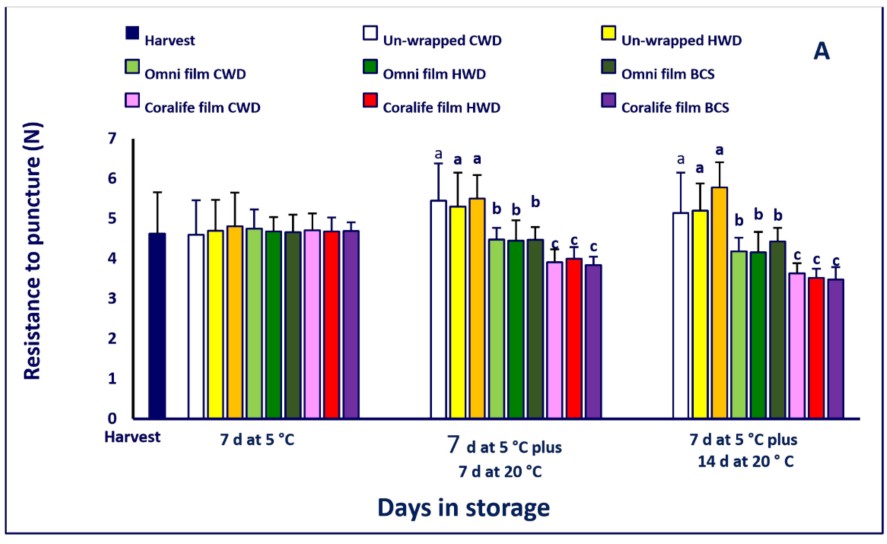

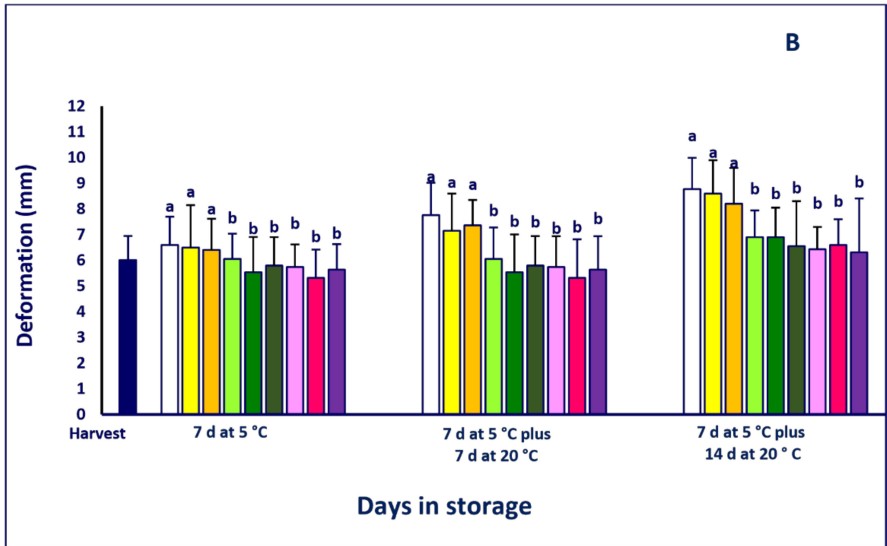

**Figure 4.** Changes in Resistance to puncture (**A**) and deformation (**B**) mness in 'Simeto', as affected by fungicidal treatments and packaging. For each storage period, means followed by different letters are significantly different at $p \leq 0.05$. Means separation was accomplished by Duncan's multiple range test. Vertical bars are the standard deviations of (n = 10).

Fruit deformation changes with storage time and film packaging (data not shown). As a general trend, it increases with storage time in all treatments, but the changes increase in the following order: Coralife packages, Omni packages, and unpackaged fruit (Figure 4B).

All color components (L*, chroma, and hue angle) are affected by all experimental factors (data not shown). The L* values decrease in all treatments with storage, with the highest reductions occurring in all treatment combinations with SBC, especially in unpackaged and Omni-packaged fruit (Figure 5A). A similar trend is detected for the chroma values (Figure 5B), and the h angle (Figure 5C), with the highest reductions occurring in unpackaged fruit, and in all treatment combinations with SBC (Figure 5B,C).

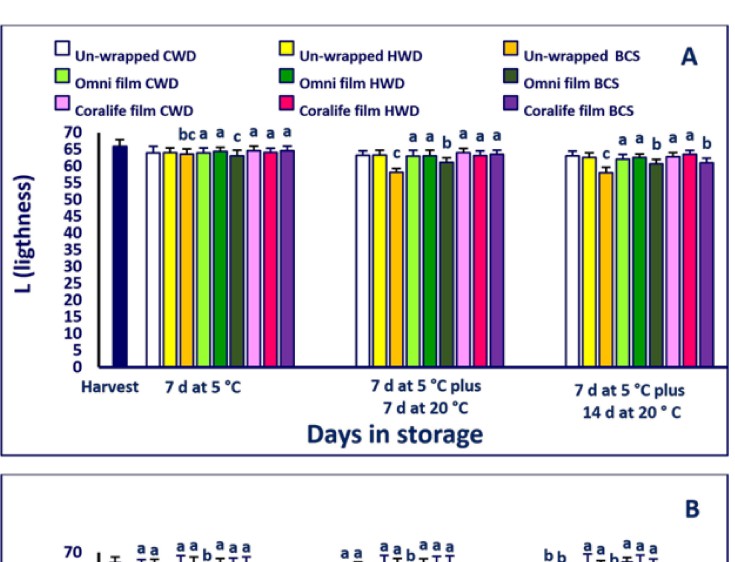

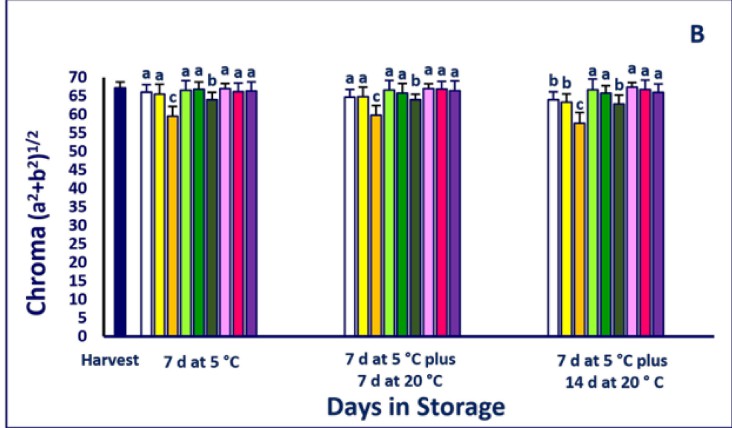

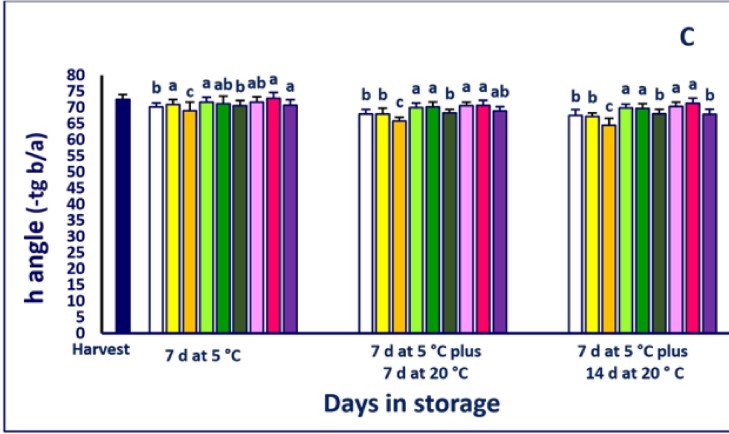

**Figure 5.** Peel color components L* (**A**), chroma (**B**), and hue (**C**) of 'Simeto' mandarins as affected by fungicidal treatment and packaging. For each storage time, means separation was accomplished by Duncan's multiple range test. Means followed by different letters are significantly different at $p < 0.05$. Vertical bars indicate the standard deviation (n = 10).

### 3.4. Juice Chemical Composition

All juice chemical parameters are affected by storage time. Some of them are also affected by packaging, and the interaction between packaging and the storage time (pH, citric acid, TSS, ethanol, and acetaldehyde); the fungicidal treatments have no significant effect (Tables 3 and 4). The pH increases with time in all treatments, but at slower rates in unpackaged and Omni-packaged fruit (Table 3). Citric acid content declines with storage in all treatments, in an increasing rate according to the following order: unpackaged, Omni packages, and Coralife packages (Table 3). TSS undergoes slight changes, and differences among packaging treatments are negligible, although a slight increasing trend occurs in unpackaged and Omni-packaged fruit; an opposite trend is detected in Coralife packages (Table 3). Vitamin C is stable over the whole storage time, and no significant difference is detected among treatments (Table 3).

**Table 3.** Juice pH, citric acid, total soluble solids (TSS), and vitamin C content of 'Simeto' mandarins after seven days at 5 °C and 85–90% RH, plus seven or fourteen days at 20 °C and 55–60% RH.

| Storage Period | pH | Citric Acid (g $L^{-1}$) | TSS (%) | Vitamin C (mg $L^{-1}$) |
|---|---|---|---|---|
| Harvest | 3.46 ± 0.072 | 12.0 ± 0.017 | 11.63 ± 0.16 | 321.2 ± 6.9 |
| 7 d at 5 °C | | | | |
| Unwrapped | 3.53 ± 0.028 b [1] | 11.0 ± 0.29 a | 11.58 ± 0.20 b | 323 ± 2.4 a |
| Omni film | 3.52 ± 0.012 b | 10.5 ± 0.20 b | 11.82 ± 0.13 a | 321 ± 3.2 a |
| Coralife film | 3.64 ± 0.029 a | 9.40 ± 0.27 c | 11.27 ± 0.17 c | 324 ± 3.8 a |
| 7 d at 5 °C plus 7 d at 20 °C | | | | |
| Unwrapped | 3.63 ± 0.054 b | 10 ± 0.17 a | 11.98 ± 0.24 a | 326 ± 6.3 a |
| Omni film | 3.61 ± 0.019 b | 9.6 ± 0.01 b | 11.67 ± 0.12 b | 319 ± 3.7 a |
| Coralife film | 3.81 ± 0.016 a | 8.6 ± 0.15 c | 11.23 ± 0.17 c | 325 ± 4.4 a |
| 7 d at 5 °C plus 14 d at 20 °C | | | | |
| Unwrapped | 3.70 ± 0.055 b | 9.6 ± 0.39 a | 12.2 ± 0.21 a | 334 ± 11.1 a |
| Omni film | 3.77 ± 0.037 b | 9.1 ± 0.37 b | 12.1 ± 0.24 a | 329 ± 9.2 a |
| Coralife film | 3.89 ± 0.028 a | 7.8 ± 0.34 c | 11.2 ± 0.22 b | 322 ± 9.4 a |
| ANOVA | | | | |
| Source | df | F-ratio | F-ratio | F-ratio | F-ratio |
| Period (P) | 3 | 7.37 *** | 645 *** | 9.56 *** | 1.67 ns |
| Treatment (T) | 2 | 0.61 ns | 0.37 ns | 0.15 ns | 0.68 ns |
| Film (F) | 2 | 3.70 * | 0.182 *** | 72.5 *** | 0.43 ns |
| P × T | 6 | 0.68 ns | 0.38 ns | 0.96 ns | 0.14 ns |
| P × F | 6 | 0.63 ns | 20.8 *** | 14.2 *** | 0.16 ns |
| T × F | 4 | 0.53 ns | 0.38 ns | 0.17 ns | 0.11 ns |
| P × T × F | 12 | 0.52 ns | 0.12 ns | 0.18 ns | 0.28 ns |

[1] Means in columns for each storage period followed by different letters are significantly different at the 5% level, using the Duncan's multiple range test. ns, *, ***, = non-significant, significant at $p \le 0.05$, significant at $p \le 0.001$, respectively.

Total phenols compounds increase slightly with storage; differences among treatments are never significant (Table 4). An opposite trend is observed for the antioxidant activity, which significantly decreases with storage. Significant differences among packages are detected at the end CS, when the TEAC values of the unpackage treatment are lower than Coralife fruit (Table 4).

Juice ethanol increases with storage, with final levels noticeably higher than at harvest time (Table 4). However, while the increases in unpackaged and Omni-packaged fruit are rather moderate at end of storage (2.1 and 1.7 fold, respectively), in Coralife-packaged fruit, the final ethanol content is 2.6 times higher than harvest time.

Acetaldehyde shows the same trend as ethanol: it increases with storage in all treatments, but at a lower and similar rate in unpackaged and in Omni-packaged fruit, compared to Coralife fruit (Table 4).

**Table 4.** Juice total phenols, antioxidant activity (TEAC), ethanol, and acetaldehyde content of 'Simeto' mandarins after seven days at 5 °C and 85–90% RH, plus seven or fourteen days at 20 °C and 55–60% RH.

| Storage Period | Total Phenols (mg L$^{-1}$) | | TEAC (mmol Trolox eq. L$^{-1}$) | Ethanol (mg L$^{-1}$) | Acetaldehyde (mg L$^{-1}$) |
|---|---|---|---|---|---|
| Harvest | 520.6 ± 19.7 | | 25.2 ± 0.09 | 39.2 ± 2.53 | 2.8 ± 0.30 |
| 7 d at 5 °C | | | | | |
| Unwrapped | 529.2 ± 26.6 a [1] | | 24.8 ± 0.83 b | 42.8 ± 2.68 b | 3.23 ± 0.47 b |
| Omni film | 526.6 ± 18.2 a | | 25.4 ± 0.62 a,b | 42.3 ± 2.34 b | 3.63 ± 0.40 b |
| Coralife film | 523.5 ± 22.7 a | | 25.9 ± 0.76 a | 47.4 ± 5.59 a | 4.35 ± 0.70 a |
| 7 d at 5 °C plus 7 d at 20 °C | | | | | |
| Unwrapped | 538.4 ± 27.2 a | | 24.2 ± 0.54 a | 62.4 ± 11.5 b | 4.43 ± 0.77 b |
| Omni film | 532.3 ± 9.61 a | | 24.8 ± 0.75 a | 66.0 ± 5.96 b | 4.35 ± 0.54 b |
| Coralife film | 526.6 ± 28.6 a | | 23.6 ± 0.98 a | 75.2 ± 4.74 a | 7.78 ± 0.99 a |
| 7 d at 5 °C plus 14 d at 20 °C | | | | | |
| Unwrapped | 568.7 ± 22.3 a | | 23.9 ± 2.98 a | 80.8 ± 6.46 b | 7.58 ± 0.68 b |
| Omni film | 565.4 ± 12.1 a | | 22.1 ± 1.28 a | 69.3 ± 26.2 b | 6.90 ± 0.77 b |
| Coralife film | 532.6 ± 32.2 a | | 20.3 ± 2.12 a | 102.8 ± 11.4 a | 9.18 ± 1.21 a |
| ANOVA | | | | | |
| Source | df | F-ratio | F-ratio | F-ratio | F-ratio |
| Period (P) | 3 | 15.9 *** | 18.1 *** | 114.6 *** | 236.1 *** |
| Treatment (T) | 2 | 0.01 ns | 0.10 ns | 0.16 ns | 0.82 ns |
| Film (F) | 2 | 0.44 ns | 0.58 ns | 13.9 *** | 51.2 *** |
| P × T | 6 | 0.01 ns | 0.10 ns | 0.86 ns | 0.16 ns |
| P × F | 6 | 0.09 ns | 1.04 ns | 5.11 ** | 11.8 *** |
| T × F | 4 | 0.003 ns | 0.03 ns | 0.66 ns | 0.28 ns |
| P × T × F | 12 | 0.002 ns | 0.12 ns | 0.42 ns | 0.11 ns |

[1] Means in columns for each storage period followed by different letters are significantly different at the 5% level, using the Duncan's multiple range test. ns, **, ***, = non-significant, significant at $p \leq 0.01$, significant at $p \leq 0.001$, respectively.

### 3.5. Sensory Evaluation

The sensory analysis reflects, in part, the results of the objective analysis. Crunchiness decreases very rapidly in unwrapped and Omni-packaged fruit, although at a slower rate; in contrast Coralife fruit experiences slight changes in crunchiness, even after 14 d of SMC (Table 5).

**Table 5.** Sensory evaluation of 'Simeto' mandarins after seven days at 5 °C and 85–90% RH, plus seven or fourteen days at 20 °C and 55–60% RH.

| Storage Period | Crunchiness | Off-Flavor | Off-odor | Acceptability |
|---|---|---|---|---|
| Harvest | 9 ± 0 | 1 ± 0 | 1 ± 0 | 9 ± 0 |
| 7 d at 5 °C | | | | |
| Unwrapped | 6.5 ± 0.74 c [1] | 2.2 ± 0.59 a | 1.3 ± 0.46 b | 5.9 ± 0.83 c |
| Omni film | 7.2 ± 0.86 b | 1.5 ± 0.51 b | 1.8 ± 0.68 a | 7.0 ± 0.92 b |
| Coralife film | 8.9 ± 0.26 a | 2.6 ± 0.63 a | 1.7 ± 0.46 a | 8.0 ± 0.65 a |
| 7 d at 5 °C plus 7 d at 20 °C | | | | |
| Unwrapped | 5.0 ± 0.76 c | 2.7 ± 0.88 a | 2.8 ± 0.68 b | 4.5 ± 1.40 c |
| Omni film | 6.5 ± 0.51 b | 1.7 ± 0.46 b | 3.0 ± 0.76 b | 5.7 ± 0.61 b |
| Coralife film | 8.5 ± 0.64 a | 2.8 ± 0.67 a | 3.9 ± 0.70 a | 7.6 ± 0.76 a |
| 7 d at 5 °C plus 14 d at 20 °C | | | | |
| Unwrapped | 1.9 ± 0.83 c | 4.3 ± 0.90 a | 3.2 ± 0.80 b | 2.9 ± 0.91 c |
| Omni film | 5.3 ± 0.72 b | 2.8 ± 0.73 b | 3.8 ± 0.86 a | 5.4 ± 0.63 b |
| Coralife film | 8.1 ± 0.70 a | 4.9 ± 0.88 a | 3.9 ± 0.80 a | 6.9 ± 0.91 a |

**Table 5.** *Cont.*

| Storage Period | | Crunchiness | | Off-Flavor | Off-odor | Acceptability |
|---|---|---|---|---|---|---|
| | | | A N O V A | | | |
| Source | df | F-ratio | | F-ratio | F-ratio | F-ratio |
| Period (P) | 3 | 116 *** | | 131 *** | 184 *** | 239 *** |
| Treatment (T) | 2 | 0.72 ns | | 0.27 ns | 0.99 ns | 0.91 ns |
| Film (F) | 2 | 394 *** | | 37.8 *** | 14.3 *** | 113 *** |
| P × T | 6 | 0.12 ns | | 0.17 ns | 1.08 ns | 0.64 ns |
| P × F | 6 | 24.3 *** | | 6.00 *** | 3.52 ** | 16.2 *** |
| T × F | 4 | 0.24 ns | | 0.56 ns | 0.77 ns | 0.21 ns |
| P × T × F | 12 | 0.65 ns | | 1.19 ns | 0.86 ns | 0.80 ns |

[1] Means in columns for each storage period followed by different letters are significantly different at the 5% level, using the Duncan's multiple range test. ns, **, ***, = non-significant, significant at $p \leq 0.01$, significant at $p \leq 0.001$, respectively.

Off-flavor increased with storage in all treatments, especially in unwrapped and Coralife fruit, which at the end of storage are scored 4.3 and 4.9, respectively, against 2.8 for Omni fruit (Table 5). Even off-odor increases with storage, but in this case, differences among treatments are less marked than off-flavor, although Omni and Coralife fruit always received higher scores (Table 5).

Overall acceptability declines in all treatments with storage, particularly in unpackaged fruit, which after 7 d of SMC are scored as below the limit of acceptability. Acceptability also declines rapidly in Omni fruit, although the score is always above the limit of acceptability. In contrast, acceptability declines slowly in Coralife fruit, which, even after 14 d of SMC, received an average score of 6.9 (Table 5).

## 4. Discussion

The two polymeric films, as well as SBC, had a distinct impact on post-harvest physiology and behavior of 'Simeto' mandarins, as discussed below.

The respiratory activity ($CO_2$ flux) measured immediately after film removal is approximately 4–5 fold higher in Coralife fruit than in Omni fruit. This difference underlines the higher resistance to $CO_2$ diffusion of Coralife film compared to Omni film. Since the film resistance to gas diffusion acts in cohesion with the fruit peel resistance, a high film barrier to gases implies high levels of in-package $CO_2$, and, in turn, even higher $CO_{2epp}$ [33,34]. Indeed, according to the barrier properties of the films, results of in-package gas composition only in part agreed with the expected values. In Omni packages held at 20 °C, in-package $CO_2$ partial pressure is approximately 3-fold lower than the expected values reported in Table 2, while $O_2$ in-package partial pressure is approximately 6 kPa higher than the expected values reported in Table 2. Although at a lower extent, even in Coralife packages, both $CO_2$ and $O_2$ partial pressures are, respectively, lower and higher than expected. This divergence is due in part to film sealing defects, but also to a respiratory activity lower than that considered when the packages were designed. In this study, overall respiration gradually decreases with storage, both in unpackaged fruit and in packaged fruit, when measurements were taken 5 h after the packages were opened. A decrease in respiration normally occurs in most fruit and vegetables after harvest, apart from the climacteric rise of some fruits, or in cases of incipient microorganism infections [35]. Different factors may be responsible for this decline, such as temperature and ambient gas composition. A decline in respiration can also occur from the depletion of respirable substrates, but this event is more likely to occur in leafy vegetables [35], which are poor of respirable substrate, rather than in fruit rich in sugars or starch [35]. In citrus fruit, a major role is played by the increase in resistance to gases diffusion consequent to peel dehydration; a higher barrier to gas diffusion would increase the gradient of $O_2$ and $CO_2$ partial pressures between fruit tissues and the atmosphere [36].

Our results of unwrapped fruit, and in part of Omni-packaged fruit, which despite film wrapping lost relevant weight, agree with previous studies [37–39]. In contrast, in Coralife fruit, the decline of respiration detected 5 h following film removal could in part be consequent to a reduction of available substrate. In fact, weight loss in Coralife fruit was negligible, as were changes in $RCO_2$, whereas the dramatic increase in $CO_{2epp}$, and the marked decline in $O_{2epp}$ while fruit were packaged, could shift, in part, the aerobic metabolism to the anaerobic metabolism. This hypothesis is justified by the fact that in Coralife fruit, TSS, but even more so, citric acid content decreases at a faster rate than in the other treatments, while both acetaldehyde and ethanol are always higher.

An increase in $CO_{2epp}$, and consequently a reduction of $O_{2epp}$, caused by peel dehydration was previously reported in citrus fruit [37,38,40,41].

Both polymeric films barely affected the headspace temperature but have a strong effect on in-package humidity, particularly the Coralife packages (average RH $\approx$ 99%), where even slight fluctuations of environment or in-package temperature were sufficient to cause condensation. In contrast, the in-package humidity level of Omni packages led to a transient appearance of condensation only when fruit from CS were moved to SMC.

The level of humidity surrounding the fruit, and the presence of condensation on fruit surface, has an outstanding effect on decay (almost completely due to P. digitatum), weight loss, overall appearance, and overall percentage of marketable fruit. In unpackaged fruit, decay is very low and barely conditioned by storage time or fungicidal treatments; infections mostly developed in the first days of storage, and thereafter, peel drying and the low environmental humidity prevented any further infections [8,9,42,43]. In contrast, in packaged fruit, ungerminated conidia at harvest time, or conidia dispersed by diseased adjacent fruit [44], could have germinated later and grown on fruit surfaces during storage, thus, increasing decay incidence [23]. In fact, losses are markedly higher in Coralife packages, where the water saturated headspace, alongside with condensation, created more favorable conditions for pathogens.

SBC, just as other bi- and carbonate salts, has shown the capability to control decay in citrus fruit, particularly Penicillium digitatum and P. italicum, although their effectiveness has not been always consistent, depending on a number of factors, such as the infection stage, the fruit maturity, the species, and the treatments conditions [8,9,20,45–47]. In this study, the presence of dissolved salt of SBC spread on fruit surface, is very effective in preventing pathogens' growth. The high capacity of SBC to prevent decay is particularly evident in Coralife-packaged fruit, where the water vapor saturated head-pace, and the presence of condensation on fruit surfaces caused high losses due to decay in CWD and HWD treatments.

SBC also confirmed its negative effect on fruit freshness by enhancing transpiration and hastening ageing [20,25,48]. This negative effect was strongly dependent on the moisture content of the ambient surrounding the fruit: the higher the humidity, the lower the phytotoxic effect. As a result, freshness declines very rapidly in BCS unpackaged fruit, followed by Omni fruit. In these treatments, differences in weight loss, overall appearance, AMF, and PMF are relevant between BCS and the other two fungicidal treatments (CWD and HWD). In contrast, the overall results of the combination BCS plus Coralife film are impressive from a commercial point of view: at the end of the trial, the percentage of marketable fruit was still around 95%, about 20–25% higher than CWD-Coralife and HWD-Coralife combinations.

Contrary to previous results, in this study hot water dip does not delay or prevent pathogens' growth. This unexpected inefficacy could be due to the higher susceptibility of mandarins to pathogens, and the advanced maturity stage of the fruit at harvest, which showed a very soft peel, but also due to the relatively short duration of immersion and low water temperature, as previously reported by Palou et al. [46].

Firmness loss is an important component of quality that considerably affects easy-peeling citrus fruit during storage, and greatly influences consumers' choice. Previous studies with citrus fruit show a positive effect of film wrapping in slowing down firmness

loss over storage [1,7,40]. In this study, the reduction of resistance to puncture of the peel in packaged fruit, especially in Coralife fruit, changes significantly over storage, although is barely perceivable from the sensory point of view. In contrast with previous results, unpackaged fruit shows an increase in resistance to puncture with storage. This apparently strange result, previously reported by D'Aquino et al. [8,9], was the response to the severe water loss experienced by the peel of unwrapped fruit, which acquired a leathery consistence. Nevertheless, the leather consistency of the peel does not affect the softening processes of the endocarp tissues, which, as shown by result of the deformation test, markedly increases in unpackaged fruit, while showing negligible changes in packaged fruit, particularly in Coralife packages, due to the high levels of in-package humidity.

Despite peel color not necessarily being associated with taste quality, it generally, markedly, drives consumer decisions [49]. Generally high values of L and chroma are typical of fresh fruit with vivid and bright colors, whereas a decrease in L and chroma, together with the hue angle, may be linked with ageing and senescence, as overall fruit color tends to be darker. Our results show a general decreasing trend of all color parameters (L, chroma and hue angle) with storage, with the highest reduction occurring in fruit treated with BCS or unpackaged fruit; the lowest changes are detected in Coralife fruit, where water loss is negligible, thus confirming how severe water loss and SBC are responsible for peel darkening and browning, whereas high humidity levels prevent color changes.

Chemical composition undergoes gradual change that mainly concern pH, citric acid, ethanol, and acetaldehyde; components that are not correlated with the healthy benefits of citrus fruit, but that are responsible for flavor and taste deterioration [50]. Nevertheless, due to the high initial level of citric acid, the loss of acidity, instead of worsening the fruit sensorial properties, improves the ratio of TSS/acidity, and, to some extent, even the eating quality.

High levels of juice ethanol and acetaldehyde are commonly used as indicators of off-flavor and bad odors in citrus fruit [51]. In general, a decrease in flavor acceptability is associated with levels of ethanol higher than $1 \, g \, L^{-1}$ [52]. Nevertheless, aroma and taste sensations result not only from the interaction between sugars and acids on one hand, and acetaldehyde and ethanol on the other, but also from the combination of several compounds, which are synthetized during storage, thus, it is not necessarily moderately high levels of ethanol or acetaldehyde that are associated with off-flavor perception [53], as our data on sensory evaluation show. The same authors [53], in accordance with Ummarat et al. [54], conclude that rather than only ethanol and ethyl esters, other volatiles affect sensory attributes in mandarins. Based on our results, a marked contribution to the decline of acceptability comes from changes in firmness, which amongst the tested sensory attributes was the variable with the highest association with loss of acceptability.

Citrus fruit are rich in phenolic compounds, vitamins, and other phytochemicals linked to a lowered risk of chronic and degenerative diseases [55,56]. For this reason, it is important that storage conditions and post-harvest treatments are planned in such a way as to minimize the impact on phytochemicals, in order to maintain their potential health benefits [57].

Vitamin C is the most popular nutraceutical compound of citrus fruit, and citrus fruits are the major contributor of dietary vitamin C to humans [58–60]. Vitamin C is thermolabile [60,61], and numerous post-harvest factors, such as storage temperature, relative humidity, mechanical stress, heat treatments, and surface coatings, but also juice pH and organic acid levels, may affect its stability and content [60–62]. Nevertheless, several studies show a stability of vitamin C, even when fruit is stored under harsh conditions. For example, HWD for 3–5 min at 40–50 °C barely affected vitamin C after several weeks of cold storage [32,63–65], whereas no, or minor changes, in vitamin C content of citrus fruit were detected after two months, or even longer storage periods, at room temperature [9,66,67]. As vitamin C is stable in acidic conditions, it is likely that the initial low pH and relatively high citric acid content of 'Simeto' mandarins favored its stability, even at warm temperature.

The antioxidant capacity shows a slight decline in storage of unwrapped fruit and of Omni fruit, while a higher decline is detected in Coralife fruit; the reduced $O_2$ partial pressure, alongside with the high levels of $CO_2$, might have accelerated the declining trend of the antioxidant capacity. Nevertheless, total phenol compounds, which in citrus, besides phenolic acids, include flavonoids [56], show a slight, but significant, overall increase, underlining an overall high stability of the nutraceutical components of 'Simeto' mandarins over the three weeks of storage at 20 °C.

## 5. Conclusions

Results of this study show that unpackaged 'Simeto' mandarins stored for 7 d at 5 °C, plus 14 d at 20 °C and 55–60% RH to simulate retail conditions and the time interval before consumption at home, aged rapidly, with most of the fruit considered un-marketable by the end of the first 7 d of SMC.

Film packaging delays the ageing process, and its efficacy increases as permeability to water vapor decreases. Thus, while the beneficial effect of Omni film is modest, that of Coralife film is outstanding. In contrast, decay incidence increases as the films' permeability to water vapor decreases.

SBC shows a high capacity to prevent decay, but becomes phytotoxic to peel tissues. Its phytotoxicity is extremely severe in unpackaged fruit, but it decreases as the humidity surrounding the packaged fruit increases. In terms of appearance and decay incidence, the best results are achieved when SBC is combined with the Coralife film, which prolonged fruit freshness for the entire storage time (7 d CS, plus 14 d SMC), with a negligible incidence of rotten fruit.

Although Omni and Coralife films affected fruit respiration, and in-package and endogenous $CO_2$ and $O_2$ partial pressures, in different ways, the changes in juice chemical composition, and sensory properties among fruit packaged using the two different films, or unpackaged fruit, are not substantial.

Thus, combining properly designed modified atmosphere packaging with SBC treatments could be a feasible strategy to prolong the post-harvest life of very perishable citrus fruit, and control decay by the application of a compound safe for the environment and for human health.

**Author Contributions:** Conceptualization, S.D.; data curation and validation, S.D. and A.P.; methodology, S.D.; formal analysis, S.D. and A.P.; writing—original draft preparation, S.D. and A.P.; writing—review and editing, A.G. and M.C.S. All authors have read and agreed to the published version of the manuscript.

**Funding:** This research received no external funding.

**Institutional Review Board Statement:** Not applicable.

**Informed Consent Statement:** Not applicable.

**Data Availability Statement:** Not applicable.

**Acknowledgments:** The authors gratefully wish to thank Stefano Tagliabue of Corapack (Brenna, Como) and Domenico Mura, for providing the Coralife film and technical assistance.

**Conflicts of Interest:** The authors declare no conflict of interest.

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
