# Peer review of "Decay Incidence and Quality Changes of Film Packaged ‘Simeto’ Mandarins Treated with Sodium Bicarbonate"

_horticulturae, doi:10.3390/horticulturae8050354_

Round 1

Reviewer 1 Report

- Title using "wrapped" seemed to mislead the content of this study using bag. 

  • Abstract didn't show strong findings, this can be improved. I think it should be focusing on which treatment gave the best results. If packaging is the main focus of this study, it should be emphasize in the abstract.
  • The authors should try to link different factors on quality and shelf life. 

Author Response

Dear Ms. Lydia Han,

I thank a lot the reviewers for the whole work done and the time spent to review this manuscript. I hope that with their valuable revision and suggestions the revised manuscript is now worth for publication in HORTICULTURAE JOURNAL.    

Below the answer to reviewers are reported

Ref. # 1

In the title the word “wrapped” was replaced by the word “packaged”.

Both the abstract and the conclusions have been completely rewritten.

Ref. # 2

The text was checked, and quotations were corrected.

The ANOVA Table was transferred to the supplementary data part of the manuscript.

Line 499 – As suggested the sentence was clarified.

Line 510. A reference was added.

Line 517. The sentence was removed not being supported by relevant findings.

The conclusions were re-written.   

Ref. # 3

The text was checked, and quotations were corrected.

The judges who assessed the samples were three trained technicians.

In the last five years a relevant number of articles have been published on the use of compounds other than synthetic fungicides to control postharvest decay of fresh fruit including citrus fruit, but the number of articles related to packaging of citrus fruit is quite poor. Anyway, where possible recent literature citations were added.    

Thank you very much

Salvatore D’Aquino

Reviewer 2 Report

D’Aquino et al presented their results on the use of sodium bicarbonate and coatings of citrus fruits. The article is well written and appropriate for Horticulturae, thus it can be accepted for publication after a minor revision.

Line 41;43 correction of citation typos (and possibly throughout the manuscript)

The Anova table should be transferred to the supplementary data part of the manuscript

Line 499 please clarify and elaborate

Line 510 reinforce the argument by providing a reference

Line 517 the argument is not supported with relevant findings, please revise

Author Response

(The authors gave the same response as above.)

Reviewer 3 Report

Manuscrpit "Decay incidence and quality changes of film wrapped 'Simeto' mandarins treated with sodium bicarbonate" presents interesting research results that can be implemented in horticulture. The manuscript is well written and research is well planned. Figures and tables are legible. The manuscript can be published with some minor corrections.

Detailed comments:

Quoting in the text should be improved - often, in addition to numbers, there are surnames, e.g. line 37 or 43.

Please, record the units in accordance with the journal's guidelines.

line 165 - how many judges assessed the samples, were they trained judges or not?

Newer literature items (from the last 5 years) should be added to the discussion. 

Author Response

(The authors gave the same response as above.)
